# Interface potential-induced natural antioxidant mimic system for the treatment of Alzheimer's disease
Kangning Liu[1], Qi Ding[1], Doudou Cao[1], Enpeng Xi[1], Yun Zhao[1], Nan Gao [1]✉, Yajie Yang[2]✉ & Ye Yuan[1]

Although the pathogenesis of Alzheimer's disease (AD) is still unknown, the molecular pathological phenomena is clear, mainly due to mitochondrial dysfunction and central nervous system inflammation caused by imbalanced antioxidant capacity and synaptic dysfunction, so antioxidant therapy is still the preferred treatment for AD. However, although antioxidant enzymes have high catalytic efficiency, the substrate spectrum is narrow; Antioxidants have wider range of effects, but their efficiency is low. Since the antioxidant defense system in high-grade organisms is composed of both enzymatic and non-enzymatic systems, therefore we synthesized a metal-organic framework (MOF) with superoxide dismutase activity, and depending on the interface potential effect, curcumin was loaded to construct a synergistic antioxidant treatment system. More importantly, due to the complementary surface electrostatic potential between MOF and curcumin, the system exhibited both good antioxidant activity and efficient β-amyloid plaque scavenging ability, which slowed down the cognitive dysfunction in the brain of AD mice.

Alzheimer's disease (AD), as the leading cause of disability and death in the elderly, accounting for 60–80% of the dementia population, is a multifactorial, progressive, neurodegenerative disease usually characterized by memory loss, personality changes, and overall cognitive decline[1,2]. AD is now a major global health problem, with ~50 million people suffering from it. Although the pathogenesis of AD is still unknown, the molecular biology pathological phenomena are clear, mainly due to mitochondrial dysfunction and central nervous system inflammation caused by imbalanced antioxidant capacity in nerve cells, as well as subsequent synaptic dysfunction. These pathological phenomena play a key role in the pathogenesis of AD. Therefore, antioxidant therapy remains the preferred treatment for AD in clinical practice[3–5].

Based on the above pathological phenomena, antioxidant therapy strategies have been widely applied in clinical practice and are considered an important means of combating AD[6–8]. However, existing antioxidant therapy methods, whether using antioxidant enzymes alone or antioxidant small molecules, all have limitations. These limitations stem from the single mechanism of action, insufficient bioavailability, and difficulty in penetrating the blood-brain barrier of antioxidants used alone[9,10].

Considering that the antioxidant defense system in the human body is a complex network constructed by both enzymatic and non-enzymatic systems, using a synergistic antioxidant system is more effective than using a single antioxidant[11,12]. In the antioxidant defense system of advanced biology, the enzyme system mainly includes superoxide dismutase (SOD), catalase, and glutathione peroxidase (GSH-Px). The exogenous antioxidants are mainly formed by hydrophilic free radical scavengers (VC and glutathione) and lipophilic free radical scavengers (such as tocopherols, flavonoids, carotenoids, and so on). The enzymatic and nonenzymatic antioxidant systems work together to maintain or re-establish the biological redox equilibrium[13,14].

Therefore, considering the interaction between carriers and drugs, which plays an important regulatory role in drug loading, release, and other behaviors, we constructed a synergistic antioxidant treatment system by utilizing a metal-organic framework (MOF, named ZnBTC, providing SOD enzyme activity) synthesized by 1,3,5-benzenetricarboxylic acid (H$_3$BTC) and zinc nitrate hexahydrate[15–18], and then depending on the interface potential effect, the curcumin (provides nonenzymatic antioxidant and anti-inflammatory capacity, and curcumin is also a well-known AD therapeutic agent) was loaded into the MOF[14,19–23]. More importantly, due to the complementary surface electrostatic potential between ZnBTC and curcumin, they can spontaneously form complexes and exhibit excellent synergistic antioxidant effects. This system (CUR@ZnBTC) aims to improve antioxidant performance through the synergistic effect of both antioxidant enzymes and

[1]Key Laboratory of Polyoxometalate and Reticular Material Chemistry of Ministry of Education and Faculty of Chemistry, Northeast Normal University, Changchun, China. [2]Key Laboratory of Automobile Materials of Ministry of Education & School of Materials Science and Engineering, Jilin University, Changchun, China. ✉e-mail: gaon320@nenu.edu.cn; yangyajie@jlu.edu.cn

antioxidants (Fig. 1). This study not only provides a new perspective for a deeper understanding of the pathological mechanisms of AD, but also demonstrates how to use interface interactions to construct reasonable carrier-drug complexes and achieve ideal therapeutic effects.

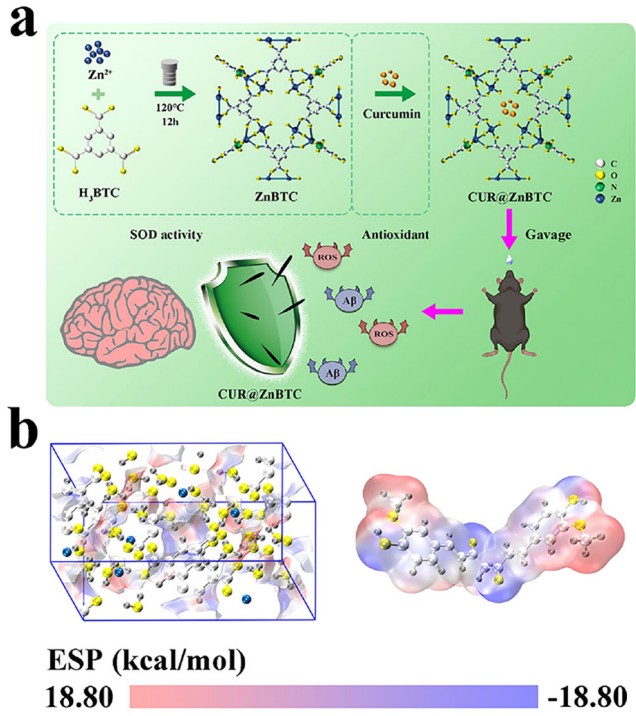

**Fig. 1 | Schematic illustration of the assembly of antioxidant system depend on interface potential and the therapeutic mechanisms. a** Schematic representation of the synthetic, assembly and therapeutic mechanisms of the CUR@ZnBTC antioxidant system for the treatment of AD. **b** Electrostatic potential surface maps of ZnBTC (left) and curcumin (right).

## Results and discussion

### Synthesis and characterization of CUR@ZnBTC

Firstly, ZnBTC was successfully synthesized using a hydrothermal synthesis technique by reacting $H_3BTC$ and zinc nitrate hexahydrate ($Zn(NO_3)_2\cdot6H_2O$) in anhydrous ethanol solvent at 120 °C (Fig. 2a). Subsequently, curcumin was loaded into ZnBTC in anhydrous ethanol solvent using solvent adsorption method to obtain CUR@ZnBTC. Scanning electron microscopy (SEM) and Transmission electron microscopy (TEM) images showed that the synthesized ZnBTC presented a rod-like morphology with a length of a few microns. And the CUR@ZnBTC nanoparticles after curcumin loading still maintained the rod-like structure and the particle size was not changed much (Fig. 2b; Fig. S1).

The XRD pattern (Fig. 2c) showed that ZnBTC has similar crystallinity to $H_3BTC$ and has good crystallinity. The successful preparation of ZnBTC was successfully confirmed by observing the unique peaks of ZnBTC at 10.1°, 17.8°, and 24.6°. After loading curcumin, CUR@ZnBTC still maintained the characteristic peaks of ZnBTC, while the characteristic peaks of curcumin could not be clearly observed, which indicated that CUR had been wrapped inside ZnBTC. From the XRD results, we can conclude that both ZnBTC and CUR@ZnBTC were successfully synthesized. In addition, the presence of CUR in ZnBTC was also confirmed by IR and UV–vis spectroscopic measurements, which further justified the conclusion drawn from our experiments (Figs. S2, S3). After that, we examined the stability of the materials in saline and bovine serum albumin, and the FTIR and XRD results showed (Figs. S4, S5) that ZnBTC and CUR@ZnBTC have good stability in salts and serum proteins.

The results of $N_2$ adsorption–desorption experiments of ZnBTC are shown in Fig. 2d. The BET specific surface area of ZnBTC was 943.421 $m^2\,g^{-1}$, the total pore volume was 0.34 cc $g^{-1}$. Its pore size was 1.356 nm, suitable for the incorporation of CUR molecules. As shown in Fig. 2e, the BET-specific surface area of CUR@ZnBTC was 468.093 $m^2\,g^{-1}$, the total pore volume was reduced to 0.17 cc $g^{-1}$; correspondingly, the pore size of CUR@ZnBTC was 0.858 nm.

Next, the thermal properties were investigated using a thermogravimetric analyzer in an $N_2$ environment from 25–800 °C (Fig. 2f). The crystal transition temperature of $H_3BTC$ was about 320 °C; after the formation of

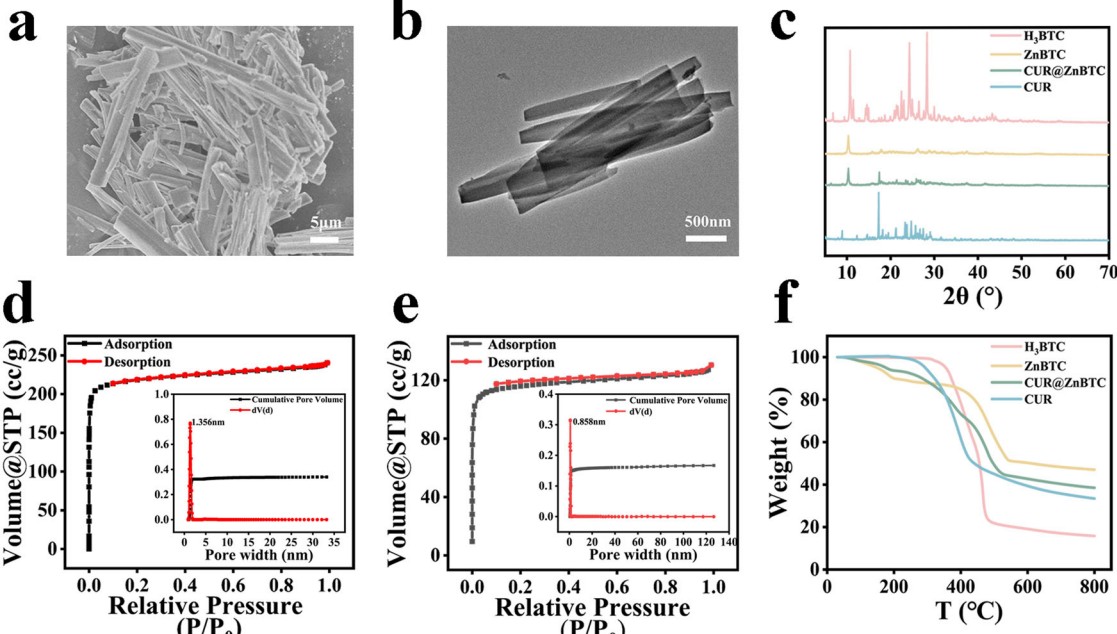

**Fig. 2 | Synthesis and characterization. a** SEM of ZnBTC; **b** TEM of CUR@ZnBTC; **c** XRD of CUR@ZnBTC; **d** Nitrogen adsorption and desorption isothermal curves of ZnBTC at 77 K (inset: average pore size distributions); **e** Nitrogen adsorption and desorption isothermal curves of CUR@ZnBTC at 77 K (inset: average pore size distributions); **f** Thermogravimetric Analysis (TGA) of CUR@ZnBTC.

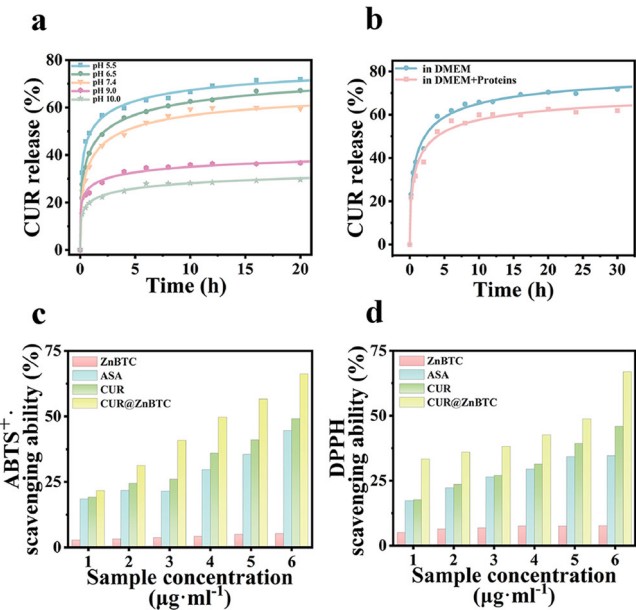

**Fig. 3 | Curcumin release and antioxidant testing experiments. a** Release curves of CUR at different pH conditions; **b** CUR release curves in Dulbecco's Modified Eagle Medium (DMEM) and simulated cellular environment (DMEM + 1 mg ml$^{-1}$ bovine serum albumin +1 mg ml$^{-1}$ acid-hydrolyzed casein); **c** ABTS radical scavenging activity; **d** DPPH· radical scavenging activity.

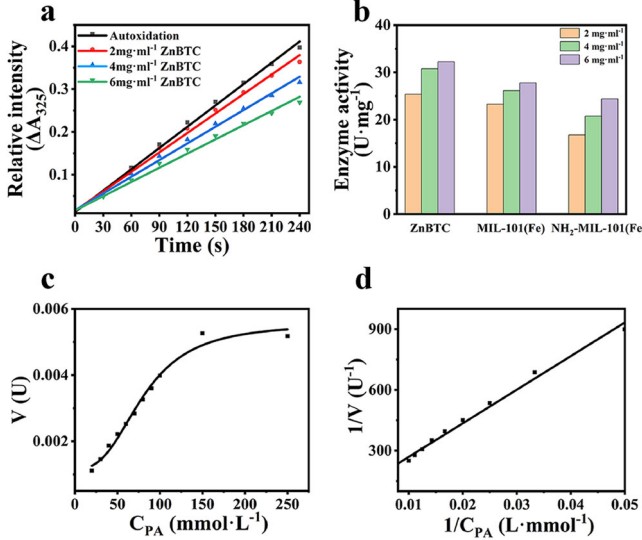

**Fig. 4 | Determine of the SOD-like activity. a** The initial pyrogallol oxidation profile inhibited by ZnBTC ; **b** Comparison of SOD enzyme activity of different materials; **c** Kinetic profiles of pyrogallol autoxidation inhibition by ZnBTC; **d** Lineweaver–Burk plots for the inhibition of pyrogallol autoxidation by ZnBTC.

MOF, there is an obvious change of the weight loss above 410 °C. Meanwhile, the weight loss of H$_3$BTC is close to 100% and that of ZnBTC is only 57%. Therefore, the increase in transition temperature and the decrease in weight loss of ZnBTC compared to H$_3$BTC indirectly confirmed the successful preparation of ZnBTC. Compared to ZnBTC, CUR@ZnBTC has a weight loss of 20% in the range of 240–400 °C, which should be attributed to the curcumin loaded in ZnBTC framework. Furthermore, the decomposition of CUR@ZnBTC was slower than that of native curcumin, indicating that the encapsulation of curcumin in ZnBTC framework could greatly improve the activity and stability of curcumin.

## Drug release study and antioxidant test of the antioxidant system

Curcumin (1,7-bis(4-hydroxy-3-methoxyphenyl)−1,6-heptanediyl-3,5-dione) is a naturally occurring polyphenol diketone extracted from the roots of herbaceous plants and has been demonstrated to be a novel naturally occurring source of therapeutic agents for AD[19]. Curcumin can directly scavenge free radicals both in vivo and in vitro by increasing SOD and GSH-Px activities, decreasing MDA content, inhibiting free radical generation and enhancing their scavenging, and alleviating oxidative stress, and can scavenge ROS faster than other polyphenolic compounds[20,21], making it a typical antioxidant. As an antioxidant and anti-inflammatory therapeutic drug, it can effectively improve the cognitive function of AD patients, reverse the neurodegeneration caused by Aβ generation, prevent β-amyloid formation, reduce β-amyloid plaques, and inhibit Aβ aggregation[22,23].

The CUR@ZnBTC antioxidant system was constructed in ethanol solvent. After the uploading experiment, the content of the remaining CUR in the supernatant was determined by UV spectrophotometry, the results showed that the uploading reached 47.31% (Fig. S6). Then, the geometry optimization and frequency calculations of ZnBTC–CUR interactions were further calculated, the results indicated that ZnBTC had a stronger adsorption capacity towards CUR (Fig. S7). Then, we investigated the ability of CUR@ZnBTC to release CUR under different pH (pH = 5.5, 6.5, 7.4, 9.0, 10.0) conditions (Fig. 3a). The results showed that under acidic conditions, the release rate and release amount of CUR were larger than under physiological condition and alkaline conditions (Fig. S8). This might be because under acidic conditions, the surface electrostatic potential of ZnBTC and CUR no longer matched, thus accelerating the release of CUR. In this work, our focus was on the release of CUR by CUR@ZnBTC under physiological conditions, and according to the release profile, CUR could achieve 61% release at pH 7.4 while releasing at an appropriate rate. This slow-release effect was related to the encapsulation of curcumin by ZnBTC and its affinity for Zn$^{2+}$, which made curcumin stay longer and more effective in the body.

Afterwards, we investigated the release properties of CUR in Dulbecco's Modified Eagle Medium (DMEM) and simulated cellular environment (DMEM + proteins) (Fig. 3b). The results showed that the release rates of CUR in both DMEM and simulated cellular environment could reach 60–70%, and the release rate in simulated cellular environment was slightly smaller than that in DMEM. To further investigate the effect of adsorbed proteins on the release of CUR, we prepared a protein- CUR@ZnBTC complex, buffered by DMEM, incubated for 1, 3, 5, 10, 20, 30, 40 h and then centrifuged to obtain supernatants, respectively. Then, the supernatants were determined by gel electrophoresis, the results indicated that even after 40 h incubation, most proteins remained in the supernatant (Fig. S9). The above experimental results demonstrated that although some proteins might adsorb on the surface of CUR@ZnBTC, it did not significantly affect the release rate and amount of CUR. The antioxidant properties of the materials were examined using DPPH radical scavenging activity test and ABTS radical scavenging activity test. The results showed that CUR has good antioxidant capacity and stronger anti-inflammatory properties than common antioxidant ascorbic acid[24,25]. The antioxidant properties of CUR@ZnBTC are greatly enhanced due to the presence of CUR and are stronger than that of single CUR (Fig. 3c, d).

The SOD activity of the material was determined using o-phenyltriol autoxidation method. The absorbance value of the sample tube at 325 nm was determined by spectrophotometer. Then, the SOD enzyme activity of the material was calculated. The results showed that the greater the concentration of ZnBTC, the greater the effect on O$_2^-$. scavenging effect, the lower the absorbance (Fig. 4a). Comparing with other MOF-based nanoenzymes, ZnBTC had strong enzyme activity and was positively correlated with the concentration (Fig. 4b). The activities of ZnBTC, MIL-101(Fe), NH$_2$-MIL-101(Fe) activities were 32.2175 U mg$^{-1}$, 27.7624 U mg$^{-1}$, 24.3896 U mg$^{-1}$, respectively, at the concentration of 6 mg ml$^{-1}$.

Then, to demonstrate the catalytic mechanism of ZnBTC, molecular dynamic simulation was employed. The results show that ZnBTC has a

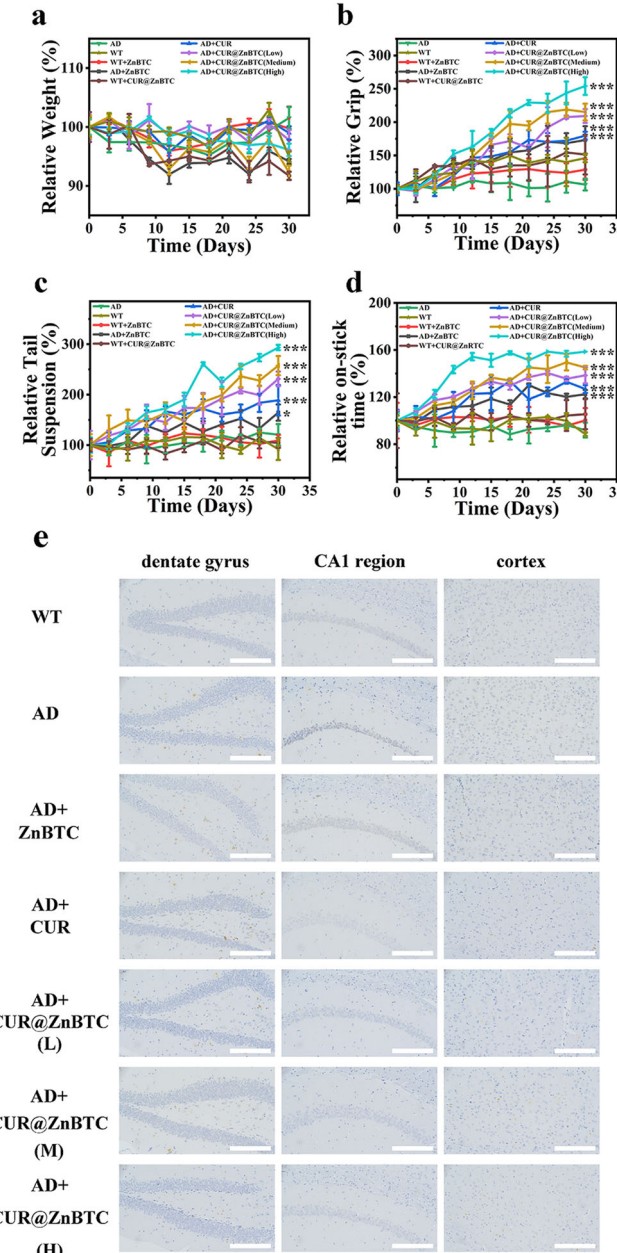

**Fig. 5 | Improvement in Aβ deposition and behavioral deficits during treatment. a** Weight change; **b** grip experiment; **c** suspended tail experiment; **d** transfer bar experiment. *$p < 0.05$, **$p < 0.01$, and ***$p < 0.001$. **e** Immunohistochemical staining images of brain sections from AD mice in different treatment groups. Scale bars are 200 μm.

IL-6. ZnBTC possesses both the good antimicrobial[16] and anti-inflammatory effects of zinc ions as well as the structural advantages of MOF[17,18]. It has good biocompatibility and is easily metabolized without causing cumulative damage. With the slow degradation of the MOF material, the long-term release of $Zn^{2+}$ can be achieved, greatly reducing the toxicity of metal ions while prolonging the functional time.

### Evaluation of the AD treatment effect of CUR@ZnBTC

Finally, we investigated the therapeutic effect of CUR@ZnBTC on AD by using 5xFAD mice. Before performing gavage, we processed the material with ultrasonic crushing. After ultrasonic crushing, the particle size of ZnBTC was 166.6 nm, the particle size of CUR@ZnBTC was 171.7 nm (Fig. S14). Hemolysis tests also prove that the material does not destroy blood cells (Fig. S15). Throughout the treatment period, we measured the body weight changes of the mice every day and observed that the body weight changes of the mice in each group were not significant, indicating that the administration of ZnBTC and CUR@ZnBTC did not have a toxic effect on the mice (Fig. 5a). To assess the sensory, cognitive and motor abilities of 5xFAD mice, we performed a series of neurobehavioral tests (Fig. 5b, c, d).

The grip experiment mainly measured the grip level and stress performance of the mice, and the test involved dragging and pulling the mice to examine their reaction ability. In the grip strength experiment, the 5xFAD mice had the worst test results. These mice were not able to respond in a timely manner when subjected to a backward pulling force on the tester, resulting in small grip force measurements. After a period of treatment, the grip strength of the 5xFAD mice was significantly enhanced, especially the most significant effect in the AD + CUR@ZnBTC group of mice. The resistance awareness of the mice can then be determined by the hanging tail experiment. This information was obtained by observing how long the mice struggled for a specific period. The mice in the AD group had the shortest swing time, and the mice in the AD + CUR, AD+ZnBTC, and AD + CUR@ZnBTC groups had the longest swing time of the struggling limb, which is consistent with the results of the grip strength experiment.

The rod-turning experiment assessed the learning and cognitive abilities of mice. The results of the experiment showed that the mice in the AD group had the worst detection results, and the mice in the AD + CUR@ZnBTC group had better detection results than those in the AD + CUR group and the AD+ZnBTC group. The mice in the AD + CUR@ZnBTC group showed more active behaviors, and the degree of activity was positively correlated with the dose. In particular, the AD + CUR@ZnBTC (high) group spent significantly more time spent on sticks than the AD + CUR@ZnBTC (low) group. The pre- and post-feeding test results of the WT mice fluctuated but did not show significant differences. These results suggest that AD + CUR@ZnBTC has a positive effect on motor coordination in mice. In conclusion, CUR@ZnBTC is effective in improving cognitive and motor abilities in 5xFAD mice. These behavioral experiments fully demonstrate the therapeutic effect of CUR@ZnBTC on AD mice by recording and measuring the behavioral responses exhibited by mice in different external environments through multiple methods to assess the cognitive responses and motor abilities of mice.

Finally, immunohistochemical assays were performed on the brains of the mice to observe the deposition of Aβ. As shown in Fig. 5e, no Aβ plaques were observed in the hippocampus of the WT group of mice, while high levels of Aβ aggregation were present in the hippocampus of the AD mice. After treatment with CUR@ZnBTC, ZnBTC, and curcumin, the number of Aβ plaques in the brains of mice was reduced to different degrees. The AD + CUR@ZnBTC group achieved a significant reduction in the Aβ content in the brains of treated 5xFAD mice with a significant reduction in the number of Aβ plaques as compared to the AD+ZnBTC and AD + CUR groups. This can be attributed to curcumin's ability to impede the formation of β-amyloid plaques, reduce the number of plaques in the brain, reduce oxidative stress, and prevent neuronal damage in the brain. The experimental results showed that long-term administration of CUR@ZnBTC can effectively reduce Aβ levels. In addition, we stained the major organs of mice

good affinity for the substrate of SOD, superoxide (Fig. S10). And the catalytic energy of the catalytic reaction is −0.27 eV, which proves that ZnBTC can spontaneously catalyze the SOD-like reaction (Fig. S11). After that, we investigated the Michaelis-Menten kinetics parameters of the materials by a double reciprocal Lineweaver-Burk plot, the results indicated that the kinetics parameters of ZnBTC was similar with the reported MOF materials (Fig. 4c, d; Figs. S12, S13; Table S1). In previous reports, ZnBTC has SOD activity and mimics endogenous enzymatic antioxidants. Where $Zn^{2+}$ is involved in regulating Cu/Zn SOD activity and indirectly regulates intracellular ROS scavenging[15]. The introduction of exogenous $Zn^{2+}$ could achieve anti-inflammatory function by restoring SOD activity. Meanwhile, $Zn^{2+}$ also inhibits the activation of NF-κB and regulates inflammatory factors. It reduces the expression of antioxidant genes SOD-1, SOD-2, and

with hematoxylin and eosin (H&E) and performed histological analyses, which showed (Figs. S16,17) that no obvious signals of organ damage and inflammatory lesions were seen. This finding further supports the safety and good tolerability of CUR@ZnBTC.

## Conclusion

In conclusion, in this work, we proposed a strategy to construct endogenous and exogenous antioxidant systems together depending on the interface potential effects for the treatment of AD, and successfully prepared CUR@ZnBTC composite nanomaterials. ZnBTC MOF exerted SOD activity and then depending on the interface potential effect, curcumin (non-enzymatic antioxidant capacity) was loaded into ZnBTC to construct a synergistic antioxidant AD treatment system, which was reflected in the improved scavenging ability of Aβ. More importantly, due to the complementary surface electrostatic potential between MOF and curcumin, they can spontaneously form complexes and exhibit excellent synergistic antioxidant effects. The experimental results showed that CUR@ZnBTC could release CUR under an AD neutral environment, and effectively scavenge ROS, alleviate oxidative stress, and inhibit the deposition of Aβ and new plaque formation. In addition, it has a strong scavenging effect on already formed plaques, which can reverse the neurological damage caused by Aβ deposition, thus improving learning and cognitive impairment in 5xFAD mice. In addition, histological analysis reveals that CUR@ZnBTC does not have any toxicity to the major organs of mice. These results provide strong evidence that CUR@ZnBTC had both good antioxidant activity and efficient Aβ plaque scavenging ability, which slowed down the cognitive dysfunction in the brain of AD mice. It is expected that this combination drug can provide a feasible strategy for the treatment of AD and provide a scientific basis for the development of more effective and safe AD therapeutic drugs in the future.

## Methods

### Basic characterization methods for materials

The KBr particles were analyzed by Fourier transform infrared spectroscopy (FTIR) in the wavelength range of $4000-400 \text{ cm}^{-1}$ using a Nicolet IS50 infrared spectrometer. SEM: tests were carried out with a JEOL-JSM-7600 scanning electron microscope instrument with an accelerating voltage of 5 kV. powder X-ray diffraction: tests were carried out with a Dmax2200PC diffractometer with a scanning range of $2-40°$ ($2\theta$) using Cu-Ka radiation, 40 kV, 200 mA, and scanning rate $5° \text{ min}^{-1}$. TEM: tests were carried out with a JEOLJ JEM-2100PLUS transmission electron microscope instrument with an accelerating voltage of 200 kV. adsorption-desorption isotherms of nitrogen were measured on a Quantachrome Autosorb-iQ2 gas adsorption instrument at a relative pressure of 0–1 bar at 77 K. Thermogravimetric analysis (TGA) was carried out on a METTLER-TOLEDO TGA/DSC$^{3+}$ thermogravimetric analyzer in the temperature range of 30–800 °C with a heating rate of $10 °C \text{ min}^{-1}$ under air conditions. The UV–vis absorption spectra of the samples were measured with a VARIAN Cary-60 UV–visible spectrophotometer in the wavelength range of 200–800 nm.

### Synthesis of ZnBTC

ZnBTC was prepared by a hydrothermal method. 1.5 g of $Zn(NO_3)_2 \cdot 6H_2O$ was dissolved homogeneously in ethanol (25 ml) by ultrasonication, and 0.63 g of $H_3BTC$ was dissolved in ethanol (35 ml), respectively. Then the two solutions were mixed homogeneously and transferred into a 100 ml stainless steel autoclave and reacted at 120 °C for 12 h. The white precipitate was collected by centrifugation and washed with ethanol three times. After vacuum drying, the final product was isolated as white powder.

### Curcumin loading

ZnBTC and CUR were dispersed in an anhydrous ethanol solution at a mass ratio of 1:2, stirred at room temperature for 24 h, and the supernatant was collected by centrifugation, and the concentration of CUR in the supernatant was determined by UV–Vis spectrophotometry, which in turn was used to calculate the amount of drug loading. The concentration of CUR in

the supernatant was determined by UV spectrophotometry. The absorbance at 425 nm was detected by UV–Vis spectrophotometer and substituted into the standard curve for calculation.

### Animal model

5xFAD mice (male, 6–8 months) were purchased from Jiangsu Jicui Pharmachem Laboratory Animal Technology Co. The rearing environment was maintained at a relatively constant temperature and humidity, with a 12 h light/dark cycle and unrestricted food and water supply. All research protocols involving animals were approved by the Animal Protection and Use Committee of Northeast Normal University. All experimental operations related to animals were in strict compliance with the "Environment and Facilities for Laboratory Animals" (GB14925-2010) "Guidelines for Ethical Review of Laboratory Animal Welfare" (GB/T 35892-2018), and the requirements of the Northeast Normal University Science and Technology Ethics Committee. All mice were acclimatized to the environment for 7 days before enrolling in the experiment.

### Statistical analysis

All experimental data were expressed as mean ± standard deviation (SD). One-way analysis of variance was used to analyze the significance between groups. Significant differences in data were analyzed by $*p < 0.05$, $**p < 0.01$, and $***p < 0.001$.

### Reporting summary

Further information on research design is available in the Nature Portfolio Reporting Summary linked to this article.

## Data availability

All data needed to support the conclusion are present in the article, supplementary information and supplementary data.

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

## Acknowledgements

This work is financially supported by the National Key R&D Program of China (Grant no. 2022YFB3805902), the National Natural Science Foundation of China (Grant nos. 22131004, U21A20330, 21975039 and 22077118), the "111" project (No. B18012), the Finance Special Project on Medical and Health Talents of the Finance Department of Jilin Province (JLSWSRCZX2023-52), WBE Liver Fibrosis Foundation (2020009) and Jilin Province Science and Technology Development Plan Project (YDZJ202401202ZYTS).

## Author contributions

N.G., Y.Yang, Y.Yuan designed the research. D.C. performed the simulations. K.L., Q.D., E.X., Y.Z. performed the experiments and analyzed the results. N.G. and Y. Yang wrote the manuscript with input from all authors.

## Competing interests
The authors declare no competing interests.
