## [Peer Review File · Communications Chemistry]

Reviewers' comments:

Reviewer #1 (Remarks to the Author):

In this manuscript, Liu et al synthesized a metal-organic framework (MOF), CUR@ZnBTC, and then loaded curcumin, a well-known AD therapeutic agent for antioxidant therapy of Alzheimer's disease (AD). The design of this manuscript was based on the combination the use of their synthesized MOF with superoxide dismutase activity and curcumin as antioxidants. The authors constructed a synergistic antioxidant AD treatment system and scavenging A β .

Recently, different types of nanomaterials with enzymatic activities have been widely reported and used in biomedicine. Several reviews on how to design and construct metal-organic framework (MOF) nanozymes have already been summarized and directed the application of MOF nanozymes in medicine: such as, Metal-Organic Framework Derived Nanozymes in Biomedicine, *Acc Chem Res.* 2020, 53(7), 1389-1400; Metal-organic framework based nanozymes: promising materials for biochemical analysis, *Chem Commun (Camb)*, 2020, 56(77), 11338-11353; Magnetic-metal organic framework (magnetic-MOF): A novel platform for enzyme immobilization and nanozyme applications, *Int J Biol Macromol.* 2018, 120(Pt B), 2293-2302 etc. Furthermore, metal-organic framework based nanozymes have been used for treatment of ROS and neuroinflammation occurred in Alzheimer's disease (AD), Parkinson's disease: examples as, Metal-Organic Framework Based Nanozyme System for NLRP3 Inflammasome-Mediated Neuroinflammatory Regulation in Parkinson's Disease. *Adv Healthc Mater.* 2023, e2303454; MOF-encapsulated nanozyme enhanced siRNA combo: Control neural stem cell differentiation and ameliorate cognitive impairments in Alzheimer's disease model. *Biomaterials.* 2020, 255, 120160; Colorimetric determination of amyloid- β peptide using MOF-derived nanozyme based on porous ZnO-Co₃O₄ nanocages. *Mikrochim Acta.* 2021, 188(2), 56; UiO-66-NH₂ Metal-Organic Framework for the Detection of Alzheimer's Biomarker A β (1-42). *ACS Appl Bio Mater.* 2024, 7(1), 182-192 etc.

As for curcumin and curcumin-based nanomedicines, they have been well studied against AD, and its ability to counteract oxidative stress and inflammation are widely used. Curcumin-Based Nanomedicines in the Treatment of Inflammatory and Immunomodulated Diseases: An Evidence-Based Comprehensive Review. *Pharmaceutics.* 2023, 15(1), 229; REVIEW: Curcumin and Alzheimer's disease. *CNS Neurosci Ther.* 2010, 16(5), 285-97; Curcumin and neurodegenerative diseases: a perspective. *Expert Opin Investig Drugs.* 2012, 21(8), 1123-40; The Mechanisms of Action of Curcumin in Alzheimer's Disease. *J Alzheimers Dis.* 2017, 58(4), 1003-1016.

It is true that CUR@ZnBTC-based curcumin nanoplatfrom in this manuscript has the antioxidant activity and β -amyloid plaque scavenging ability. However, compared with earlier reported MOF nanozymes and curcumin in the literature, the current system did not show much advantages in this specific goal. Therefore, I do not see much advance and

novelty in the current work which fulfills the criteria of publishing in Communications Chemistry. The following are some specific comments.

1. Regarding the CUR@ZnBTC enzymatic activity, the activity is rather weak. The mechanism is not clear which needs clarified. The comparison with reported MOF nanozymes is required, and related kinetic parameters have to be presented and compared.
2. More characterizations of CUR@ZnBTC loaded with curcumin should be provided. The location of curcumin and amount require further identified except inside, and the releasing studies were done too rash, different staged rates have to be reported both in vitro and in cells.
3. How the cellular proteins adsorbed on the CUR@ZnBTC and affected the releasing? This should be further characterized.
4. Regarding AD model studies, the crucial issue on how the designed nanomedicine crossed the BBB has not been clarified. More detailed experiments have to be performed and the mechanism being explained.

Reviewer #2 (Remarks to the Author):

Alzheimer disease (AD) is a complex neurodegenerative disease characterized by the mitochondrial dysfunction and central nervous system inflammation caused by imbalanced antioxidant capacity in brain nerve cells. Depending on the interface potential effect, the authors set up a complex system formed by SOD activity-like MOF and curcumin. This system demonstrates both good antioxidant activity and efficient β -amyloid plaque scavenging ability. From this point of view, I think the novelty of this work is obviously clear and should attract a broad audience. On the negative side, there are some minor points that aren't as convincing as they might be, so the authors should answer the following questions before the manuscript was accepted.

1. In Introduction section, the description of AD is inaccurate. AD is not dementia, it only accounts for about 70% of the dementia population.
2. The specific administration method of CUR@ZnBTC should be marked in Figure 1
3. The legend of the inserted figures in Fig. 2d and 2e is unclear and difficult to distinguish. The legend should clearly indicate which is ZnBTC and which is CUR@ZnBTC.
4. the figure legend of Figure 3 a and b maybe confused. In Figure 3 c and d, what does "ASA" mean?
5. In Figure 4, the "AD+CUR@ZnBTC(L), AD+CUR@ZnBTC(M), AD+CUR@ZnBTC(H)" should be written as "AD+CUR@ZnBTC(Low), AD+CUR@ZnBTC(Medium), AD+CUR@ZnBTC(High)" to make them clearer.
6. In Figure S2, legends should clearly indicate which is ZnBTC and which is CUR@ZnBTC
7. The legend of Figure S3 should be examined carefully, since there were several

mistakes.

8. Does Figure S4 mean "Solid UV/Vis spectrum of the materials"?

9. The Figure S8 was not mentioned in the main text, the authors should add it.

Reviewer #3 (Remarks to the Author):

The manuscript entitled, 'Interface Potential Induced Natural Antioxidant Mimic System for the Treatment of Alzheimer's Disease' by Liu et al. describes the synthesis of Zn based metal organic frameworks (MOFs) their antioxidant potential, loading of anti-inflammatory drug curcumin, drug release in situ and in mice model for AZ treatment. Overall this is a very interesting study, rod like MOFs are made by hydrothermal synthesis and are analyzed by TEM and SEM. The crystalline structure and shape of MOFs before and after drug loading is confirmed by XPS and SEM/TEM. The surface area and pore sizes are analyzed by nitrogen adsorption and degradation is tested by TGA. Curcumin loading reduces pore size indicating the successful loading of curcumin in MOFs. The release of drug was studied as a function of pH and time. There seems to be no significant change in drug release as a function of pH(5-6%), the legend of figure 3b should be modified to indicate the time point used for the study. Figure 3a can include a legend to correlate colors with samples and is hard to conclude the data as is. What is ASA in figure 3C? why is this a suitable control for figure 3C, 3D and 3A? Release studies and antioxidant results are not well explained in the text and should be discussed in detail. Figure 4 A-D requires legends to match the colors. This is hard to conclude any results from the figures as presented.

Other issues are as follows: Please include Curcumin loading efficacies/g or mg of MOFS? One of the key requirement of drug carriers are aqueous solubility, are MOFS prepared in this study are soluble in aqueous solution? Are physiological stability in the presence of salts and serum proteins tested for MOFS?

Response to the comments from the reviewers:

First, we would like to thank all the reviewers for their critical examinations of our manuscript. Their comments have been very helpful for revising the manuscript. We have made significant changes to address their concerns. The details of our responses are described below.

Reviewer 1:

In this manuscript, Liu et al synthesized a metal-organic framework (MOF), CUR@ZnBTC, and then loaded curcumin, a well-known AD therapeutic agent for antioxidant therapy of Alzheimer's disease (AD). The design of this manuscript was based on the combination the use of their synthesized MOF with superoxide dismutase activity and curcumin as antioxidants. The authors constructed a synergistic antioxidant AD treatment system and scavenging A β .

Recently, different types of nanomaterials with enzymatic activities have been widely reported and used in biomedicine. Several reviews on how to design and construct metal-organic framework (MOF) nanozymes have already been summarized and directed the application of MOF nanozymes in medicine: such as, Metal-Organic Framework Derived Nanozymes in Biomedicine, *Acc Chem Res.* 2020, 53(7), 1389-1400; Metal-organic framework based nanozymes: promising materials for biochemical analysis, *Chem Commun (Camb)*, 2020, 56(77), 11338-11353; Magnetic-metal organic framework (magnetic-MOF): A novel platform for enzyme immobilization and nanozyme applications, *Int J Biol Macromol.* 2018, 120(Pt B), 2293-2302 etc. Furthermore, metal-organic framework based nanozymes have been used for treatment of ROS and neuroinflammation occurred in Alzheimer's disease (AD), Parkinson's disease: examples as, Metal-Organic Framework Based Nanozyme System for NLRP3 Inflammasome-Mediated Neuroinflammatory Regulation in Parkinson's Disease. *Adv Healthc Mater.* 2023, e2303454; MOF-encapsulated nanozyme enhanced siRNA combo: Control neural stem cell differentiation and ameliorate cognitive impairments in Alzheimer's disease model. *Biomaterials.* 2020, 255, 120160; Colorimetric determination of amyloid- β peptide using MOF-derived nanozyme based on porous ZnO-Co₃O₄ nanocages. *Mikrochim Acta.* 2021, 188(2), 56; UiO-66-NH₂ Metal-Organic Framework for the Detection of Alzheimer's Biomarker A β (1-42). *ACS Appl Bio Mater.* 2024, 7(1), 182-192 etc.

As for curcumin and curcumin-based nanomedicines, they have been well studied against AD, and its ability to counteract oxidative stress and inflammation are widely used. Curcumin-Based Nanomedicines in the Treatment of Inflammatory and Immunomodulated Diseases: An Evidence-Based Comprehensive Review. *Pharmaceutics.* 2023, 15(1), 229; REVIEW: Curcumin and Alzheimer's disease. *CNS Neurosci Ther.* 2010, 16(5), 285-97; Curcumin and neurodegenerative diseases: a perspective. *Expert Opin Investig Drugs.* 2012, 21(8), 1123-40; The Mechanisms of Action of Curcumin in Alzheimer's Disease. *J Alzheimers Dis.* 2017, 58(4), 1003-1016.

It is true that CUR@ZnBTC-based curcumin nanoplatform in this manuscript has the

antioxidant activity and β -amyloid plaque scavenging ability. However, compared with earlier reported MOF nanozymes and curcumin in the literature, the current system did not show much advantages in this specific goal. Therefore, I do not see much advance and novelty in the current work which fulfills the criteria of publishing in Communications Chemistry. The following are some specific comments.

Response: Thanks for the reviewer's comments. As the reviewer pointed out, there are currently many literature reports on MOF as a carrier and nanoenzyme, as well as curcumin as a therapeutic agent for AD. However, most researchers, when choosing drug carriers, only consider the role of the carrier (transportation, or providing a certain therapeutic function) and the role of the drug (treatment), but often overlook the interaction between the carrier and the drug. The interaction between carriers and drugs plays an important regulatory role in drug loading, release, and other behaviors. Therefore, in this work, we selected representative carriers ZnBTC and drug curcumin as examples to demonstrate how to use interface interactions to construct reasonable carrier-drug complexes and achieve ideal therapeutic effects.

1. Regarding the CUR@ZnBTC enzymatic activity, the activity is rather weak. The mechanism is not clear which needs clarified. The comparison with reported MOF nanozymes is required, and related kinetic parameters have to be presented and compared.

Response: Thanks for the reviewer's comments. We compared the SOD activities of ZnBTC, MIL-101(Fe), NH₂-MIL-101(Fe) and the results showed that at a sample concentration of 2mg/ml, the activities of ZnBTC, MIL-101(Fe), NH₂-MIL-101 were 25.3713 U/mg, 23.2570 U/mg, 16.7631 U/mg; when the sample concentration was 4mg/ml, the activities of ZnBTC, MIL-101(Fe), NH₂-MIL-101(Fe) were 30.7702 U/mg, 26.1263 U/mg, 20.7274 U/mg, respectively; when the sample concentration was 6mg/ml, the activities of ZnBTC, MIL-101(Fe), NH₂-MIL-101(Fe) activities were 32.2175 U/mg, 27.7624 U/mg, 24.3896 U/mg, respectively (more details please see Fig. 4b).

Then, to demonstrate the catalytic mechanism of ZnBTC, molecular dynamic simulation was employed. The results show that ZnBTC has a good affinity for the substrate of SOD, superoxide (Fig. S10). And the catalytic energy of the catalytic reaction is -0.27 eV, which proves that ZnBTC can spontaneously catalyze the SOD-like reaction (Fig. S11).

Finally, the Michaelis-Menten kinetics parameters were determined by a double reciprocal Lineweaver-Burk plot. The results indicated that, for ZnBTC, the K_m , k_{cat} and V_{max} were $1.6015 \times 10^2 \text{ mmol} \cdot \text{L}^{-1}$, 0.0058 s^{-1} and $0.0053 \text{ mmol} \cdot \text{L}^{-1} \text{ s}^{-1}$, respectively, which also similar with the reported MOF materials (Fig. S12, S13, Table S1).

2. More characterizations of CUR@ZnBTC loaded with curcumin should be provided. The location of curcumin and amount require further identified except inside, and the releasing studies were done too rash, different staged rates have to be reported both in vitro and in cells.

Response: Thanks for the reviewer's comments. We have added data on the curcumin

loading rate in the main text, and our arithmetic formula is as follows: Drug Loading efficiency (DLC%) = Amount of curcumin in CUR@ZnBTC / Mass of CUR@ZnBTC × 100 %. When converted to mass, that curcumin loading rate is 0.89g/g ZnBTC.

For determine the location of curcumin, the geometry optimization and frequency calculations of ZnBTC-CUR interactions were further calculated, the results indicated that ZnBTC had a stronger adsorption capacity towards CUR (Fig. S7).

Finally, we investigated the release properties of CUR@ZnBTC in Dulbecco's Modified Eagle Medium (DMEM) and simulated cellular environment (DMEM + 1 mg/ml bovine serum albumin + 1 mg/ml acid-hydrolysed casein). The results showed that the release rates of CUR in both DMEM and simulated cellular environment could reach 60-70%, and the release rate in simulated cellular environment was slightly smaller than that in DMEM (Fig. 3b).

3. How the cellular proteins adsorbed on the CUR@ZnBTC and affected the releasing ? This should be further characterized.

Response: Thanks for the reviewer's comments. We prepared a 1 mg/ml bovine serum albumin + 1 mg/ml acid-hydrolysed casein + 0.05 mg/ml CUR@ZnBTC complex, buffered by DMEM, incubated for 1, 3, 5, 10, 20, 30, 40 hours and then centrifuged to obtain supernatants, respectively. Then, the supernatants were determined by gel electrophoresis, the results indicated that even after 40 hours incubation, most proteins remained in the supernatant (Fig. S9). At the same, the release rates of CUR could still reach 60-70% (Fig. 3b). The above experimental results demonstrated that although some proteins might adsorb on the surface of CUR@ZnBTC, it did not significantly affect the release rate and amount of CUR.

4. Regarding AD model studies, the crucial issue on how the designed nanomedicine crossed the BBB has not been clarified. More detailed experiments have to be performed and the mechanism being explained.

Response: Thanks for the reviewer's comments. We agree with this comment. Currently, a major impediment for development of effective therapeutic agents for AD treatment is that most small molecules or peptides are difficult to cross the blood-brain barrier (BBB). Therefore, firstly, as mentioned in our paper, we ultrasonically fragmented the material before administering gavage to the mice, after which we determined the hydrated particle size of the material, which showed that the ZnBTC was 166.6 nm, the CUR@ZnBTC was 171.7nm (Fig. S14). According to the study, nanoparticles with a diameter of less than 200 nm have been shown to cross the blood-brain barrier (Tracy, G.C., et al. Intracerebral Nanoparticle Transport Facilitated by Alzheimer Pathology and Age. *Nano Letters*. **23**, 10971-10982 (2023); Wang, X., et al. Bioinspired Adaptive Microdrugs Enhance the Chemotherapy of Malignant Glioma: Beyond Their Nanodrugs. *Advanced Materials*. **17**, e2405165 (2024); Dai, Y., et al. Nanoparticle design strategies for enhanced anticancer therapy by exploiting the tumour microenvironment. *Chemical Society Reviews*. **46**, 3830-3852 (2017)).

In addition, we also determined whether ZnBTC and CUR@ZnBTC could cross the BBB. In test group, each AD mouse was administered 20 mg of ZnBTC or

CUR@ZnBTC per kilogram of body weight by gavage once a day for 7 days. Then, at day 3, 5 and 7, the brain homogenates of WT, AD, AD+ZnBTC and AD+CUR@ZnBTC groups were analyzed for the levels of Zn as the surrogate measurement of nanomedicine levels by inductively coupled plasma mass spectrometry (ICP-MS). The experimental results show that compared to WT mice, the Zn content in the brain of AD mice is reduced, which is consistent with previous literature reports (Larry Baum, et al. Serum zinc is decreased in Alzheimer's disease and serum arsenic correlates positively with cognitive ability. *Biometals*, **23**, 173–179 (2010); Karen Cilliers. Trace element alterations in Alzheimer's disease: A review. *Clinical Anatomy*, **34**, 766–773 (2021); Jee Wook Kim, et al. Serum zinc levels and in vivo beta-amyloid deposition in the human brain. *Alzheimer's Research & Therapy*, **13**, 190 (2021)). After gavage with ZnBTC or CUR@ZnBTC, the level of Zn in the brain increased significantly (Fig. R1), which indicated that ZnBTC or CUR@ZnBTC not only crossed the BBB, but also supplemented the loss of Zn in the AD brain.

Fig. R1 Zn content in the brain of different groups of mice.

Reviewer 2:

Alzheimer disease (AD) is a complex neurodegenerative disease characterized by the mitochondrial dysfunction and central nervous system inflammation caused by imbalanced antioxidant capacity in brain nerve cells. Depending on the interface potential effect, the authors set up a complex system formed by SOD activity-like MOF and curcumin. This system demonstrates both good antioxidant activity and efficient β -amyloid plaque scavenging ability. From this point of view, I think the novelty of this work is obviously clear and should attract a broad audience. On the negative side, there are some minor points that aren't as convincing as they might be, so the authors should answer the following questions before the manuscript was accepted.

1. In Introduction section, the description of AD is inaccurate. AD is not dementia, it only accounts for about 70% of the dementia population.

Response: Thanks for the reviewer's comments. This is indeed a detail that we overlooked when reviewing the information, so we rechecked the information. According to the Chinese Alzheimer's Disease Dementia Diagnosis and Treatment Guidelines (2020 edition), dementia has become a common disease in the elderly, with Alzheimer's Disease (AD) dementia accounting for 60-80 percent of the cases, and it is the main cause of disability and death in the elderly. Now we have corrected it in the introduction.

2. The specific administration method of CUR@ZnBTC should be marked in Figure 1.

Response: Thanks for the reviewer's comments. We have added the specific mode of administration of CUR in Figure 1.

3. The legend of the inserted figures in Fig. 2d and 2e is unclear and difficult to distinguish. The legend should clearly indicate which is ZnBTC and which is CUR@ZnBTC.

Response: Thanks for the reviewer's comments. We re-wrote the legend as: “**d**) Nitrogen adsorption and desorption isothermal curves of ZnBTC at 77k (inset: average pore size distributions); **e**) Nitrogen adsorption and desorption isothermal curves of CUR@ZnBTC at 77k (inset: average pore size distributions)”

4. The figure legend of Figure 3 a and b maybe confused. In Figure 3 c and d, what does “ASA” mean?

Response: Thanks for the reviewer's comments. We have corrected the errors on the legends of Figures 3a and b. The meaning of ASA in Fig.3c and d is ascorbic acid (ASA), which has strong antioxidant properties and can be used as a control for the antioxidant properties of the materials in this paper (Gegotek, A., Skrzydlewska, E. Antioxidative and Anti-Inflammatory Activity of Ascorbic Acid. *Antioxidants*. **11**, 1993 (2022); Margaret, E. Rice. Ascorbate regulation and its neuroprotective role in the brain. *Trends Neurosci*. **23**, 209-216 (2000)).

5. In Figure 4, the “AD+CUR@ZnBTC(L), AD+CUR@ZnBTC(M), AD+CUR@ZnBTC(H)” should be written as “AD+CUR@ZnBTC(Low), AD+CUR@ZnBTC(Medium), AD+CUR@ZnBTC(High)” to make them clearer.

Response: Thanks for the reviewer's comments. We agree with this comment. We have made changes in the diagram so that it is indeed clearer and easier to understand. Due to changes in the content of the article, the new image is named Fig.5.

6. In Figure S2, legends should clearly indicate which is ZnBTC and which is CUR@ZnBTC.

Response: We would like to give our thanks to the reviewer. We have reworked the legend of

Figure S2 to make it clearer. Due to changes in the content of the article, the new image is named Fig. S1.

7. The legend of Figure S3 should be examined carefully, since there were several mistakes.

Response: Thanks for the reviewer's comments. We have rechecked the legend for Figure S3 and found several formatting errors. We have made the corrections. Thank you for your carefulness in bringing these errors in detail to our attention. Due to changes in the content of the article, the new image is named Fig. S2.

8. Does Figure S4 mean "Solid UV/Vis spectrum of the materials"?

Response: Thanks for the reviewer's comments. We agree with the reviewer's comments and we have revised the figure legend. Due to changes in the content of the article, the new image is named Fig. S3.

9. The Figure S8 was not mentioned in the main text, the authors should add it.

Response: Thanks for the reviewer's comments. We apologized for omitting it in our writing, we have added it in the main text. Due to changes in the content of the article, the new image is named Fig. S17.

Reviewer 3:

The manuscript entitled, 'Interface Potential Induced Natural Antioxidant Mimic System for the Treatment of Alzheimer's Disease' by Liu et al. describes the synthesis of Zn based metal organic frameworks (MOFs) their antioxidant potential, loading of anti-inflammatory drug curcumin, drug release in situ and in mice model for AZ treatment. Overall this is a very interesting study, rod like MOFs are made by hydrothermal synthesis and are analyzed by TEM and SEM. The crystalline structure and shape of MOFs before and after drug loading is confirmed by XPS and SEM/TEM. The surface area and pore sizes are analyzed by nitrogen adsorption and degradation is tested by TGA. Curcumin loading reduces pore size indicating the successful loading of curcumin in MOFs. The release of drug was studied as a function of pH and time. There seems to be no significant change in drug release as a function of pH(5-6%), the legend of figure 3b should be modified to indicate the time point used for the study.

Response: Thanks for the reviewer's comments. We have redone the CUR release experiments by continuously monitoring the amount of CUR in solution until the amount of CUR in the supernatant reached a constant value at each pH condition. The new release curves are shown in the new Fig.3a. We also summarized the release properties at different pH in Fig. S8.

Figure 3a can include a legend to correlate colors with samples and is hard to conclude the data as is.

Response: Thanks for the reviewer's comments. We have added legends to associate samples with colours in Fig. 3a.

What is ASA in figure 3C? why is this a suitable control for figure 3C, 3D and 3A?

Response: Thanks for the reviewer's comments. The meaning of ASA in Fig.3 is ascorbic acid (ASA), which is a natural antioxidant with strong antioxidant properties, and it is used as a control to show the strength of the antioxidant properties of the materials in this study (Gegotek, A., Skrzydlewska, E. Antioxidative and Anti-Inflammatory Activity of Ascorbic Acid. *Antioxidants*. **11**, 1993 (2022); Margaret, E. Rice. Ascorbate regulation and its neuroprotective role in the brain. *Trends Neurosci*. **23**, 209-216 (2000)).

Release studies and antioxidant results are not well explained in the text and should be discussed in detail.

Response: We would like to give our thanks to the reviewer. We agree with this comment. We have re-examined and explained the drug release experiments and antioxidant property experiments in detail in the main text.

Figure 4 A-D requires legends to match the colors. This is hard to conclude any results from the figures as presented.

Response: Thanks for the reviewer's comments. We have included legends in Figures 4A-D to match the different line colours to make the experimental results clearer. Due to changes in the content of the article, the new image is named Fig.5.

Other issues are as follows: Please include Curcumin loading efficacies/g or mg of MOFS?

Response: Thanks for the reviewer's comments. We have added data on the curcumin loading rate in the main text, and our arithmetic formula is as follows: Drug Loading efficiency (DLC%) = Amount of curcumin in CUR@ZnBTC / Mass of CUR@ZnBTC × 100%. When converted to mass, that curcumin loading rate is 0.89 g/g ZnBTC.

One of the key requirement of drug carriers are aqueous solubility, are MOFS prepared in this

study are soluble in aqueous solution?

Response: Thanks for the reviewer's comments. We agree with this comment. For most MOFs materials, or even most nano materials, they are all water dispersible (Q. Wei, et al. Hierarchically Structured and Highly Dispersible MOF Nanozymes Combining Self-Assembly and Biomineralization for Sensitive and Persistent Chemiluminescence Immunoassay. *ACS Appl. Mater. Interfaces* **2023**, *15*, 42404–42412; A. Deep, et al. Highly sensitive detection of dipicolinic acid with a water-dispersible terbium-metal organic framework. *Biosensors and Bioelectronics* **2016**, *86*, 799–804; J. Li, et al. A high-flux mixed matrix nanofiltration membrane with highly water dispersible MOF crystallites as filler. *Journal of Membrane Science*, **2019**, *591*, 117360). For our ZnBTC, it has the same property, which means that under physiological conditions, simple ultrasonication can form a good water dispersion state.

Are physiological stability in the presence of salts and serum proteins tested for MOFs?

Response: We would like to especially thank the reviewers. We added the stability experiments in the presence of salts and serum albumin. ZnBTC and CUR@ZnBTC were incubated in saline and bovine serum albumin for 3 days, and then the FT-IR and XRD of the materials were measured to determine if the structure of the materials was damaged. Figures have been added in the Supporting Information. The FTIR results showed that the peaks and positions of the material did not change (Fig. S4). The XRD results showed that the characteristic peaks of the material were still present (Fig. S5). The above results show that the material has good stability in salts and serum proteins.

REVIEWERS' COMMENTS:

Reviewer #2 (Remarks to the Author):

The authors have addressed all the issues that the reviewers put forward. I could recommend its publication in Communication Chemistry.

Reviewer #3 (Remarks to the Author):

The authors have made the requested changes. The manuscript seems suitable for publication in the present form.